# Vertical Profiling of Canadian Wildfire Smoke in the Baltimore-Washington Corridor - Interactions with the Planetary Boundary Layer and Impact on Surface Air Quality

5 Nakul N. Karle, Ricardo K. Sakai, Rocio D. Rossi, Adrian Flores, Sen Chiao

NOAA Cooperative Science Center for Atmospheric Sciences and Meteorology (NCAS-M), Howard University, Washington D.C., 20059, USA

Correspondence to: Nakul N. Karle (nakul.karle@howard.edu)

Abstract: The 2023 Canadian wildfires yielded record-breaking emissions that were transported long distances over large sections of the mid-Atlantic region, significantly impacting regional surface air quality. In this study, we analyzed the effect of long-distance transported wildfire smoke on the Baltimore-Washington Corridor (BWC), a highly populated and industrialized metropolitan region prone to air quality exceedances. Central to the analysis is the Vaisala CL61 ceilometer in Beltsville (suburban BWC), whose *linear depolarization ratio* (LDR) profiles provide a continuous, altitude-resolved fingerprint for distinguishing wildfire smoke from locally generated urban aerosols. By combining the LDR-derived with satellite imagery, surface air quality observations, and NOAA HYSPLIT trajectories, we analyzed four discrete smoke events to characterize smoke's vertical distribution and interaction with the planetary boundary layer (PBL). One of the cases showed that the timing of the smoke plume's apparent lowering over the study site in relation to synoptic frontal passage was decisive in determining its impact on air quality. In contrast, those events with well-mixed smoke in the PBL during the advection-driven conditions exhibited a clear deterioration in air quality near the surface, with particulate levels exceeding the regulation threshold. The results underscore the importance of accurately representing vertical mixing in smoke forecasts and illustrate the added value of routine ceilometer LDR measurements for real-time identification of lofted smoke plumes—information not attainable from column-integrated satellite products or surface monitors alone.

#### 1 Introduction

Wildfire smoke, particularly intense plumes, which are distinct from other kinds of air pollution, represents a significant and growing contributor to episodes of poor air quality in the Mid-Atlantic region (Black et al., 2017; Dreessen et al., 2016; Jaffe et al., 2020; Yang et al., 2022). The chemical species detected in smoke from a specific wildfire occurrence are determined by various characteristics unique to the burn location, including the type of plant burned and weather conditions (Urbanski, 2013). These smoke plumes are typically rich in particulate matter (PM) and ozone precursors (Lara et al., 2022). Fine particles fall out of the atmosphere more slowly than coarse particles, dispersing farther away from the source (Kinney, 2008). Their presence substantially degrades air quality, especially in downwind areas, even those located considerable distances from the

fire source (Hung et al., 2020; Jaffe & Wigder, 2012; Liu et al., 2016; Shen et al., 2025). The summer of 2023 starkly illustrated Canadian history's most severe wildfire seasons (Jain et al., 2024). By the end of the year, almost 6,000 fires destroyed an astounding 15 million hectares of boreal forest across numerous provinces (Natural Resources Canada, 2024). The Canadian wildfires in 2023 outnumbered previous significant wildfire occurrences, including the severe 2015 fire season, which burned roughly 4 million hectares, already 60% over the 10-year normal (Natural Resources Canada, 2016). The heavy smoke produced by the 2023 Canadian wildfires crossed continental boundaries, reaching densely populated areas of the United States at an alarming rate. The hot, dry weather was a primary cause of the fire spread, with 2023 being the warmest and driest year since 1980 (Jain et al., 2024). Episodic anomalies with implications for public health, atmospheric chemistry, and regulatory compliance with air quality standards followed.

In the last few decades, we have also seen a disproportionate rise in the frequency and scale of wildfires. Historically, the boreal forest did not burn very often. When it did, it was not over an extensive area (UNEP, 2022), but presently, the boreal forest is undergoing fire activity at an abnormal rate. According to reports, wildfires are already spreading across bigger regions, leading to record-breaking emissions year after year (Jones et al., 2024). These extensive wildfires devastated local ecosystems and generated massive plumes of smoke that traversed long distances, significantly impacting air quality in regions across North America (Bruce et al., 2025). During the 2015 wildfire season, long-range smoke transport led to a doubling of volatile organic compound (VOC) concentrations and a threefold rise in Maryland's harmful reactive nitrogen species (NO<sub>x</sub> derivatives) levels. Ozone levels also increased during this period, exceeding the national recommended thresholds (Dreessen et al., 2016). In contrast, the analysis discussed ahead will show that the 2023 smoke plume exhibited behavior that diverged from projected expectations upon reaching Maryland. This deviation underscores the importance of analyzing the variability in smoke transport processes under differing atmospheric conditions, particularly concerning the timing and nature of interactions between smoke plumes and the planetary boundary layer (PBL).

The Baltimore-Washington Corridor (BWC) is a major cosmopolitan region in the densely populated mid-Atlantic region of the U.S. It is particularly vulnerable to various air pollutants because of its geographical location and high level of urbanization. While anthropogenic emissions dominate local exceedance events during the summer, long-range transport of intense smoke from events such as wildfire can significantly change background aerosol concentrations and disrupt photochemistry (Dreessen et al., 2023). Notably, the extent to which increased smoke plumes impact ground-level air quality relies heavily on their vertical distribution and interaction with the PBL (Pahlow et al., 2005; Wang et al., 2015). Smoke layers lingering in the free troposphere can stay aloft for several days. However, its intrusion into the PBL can lead to a sudden increase in surface PM levels, prompting exceedance events (O'Dell et al., 2019). For the BWC, the Canadian 2023 wildfire smoke impact was most pronounced during the first week of June, when visibility dropped significantly, and air quality index (AQI) values frequently reached "Unhealthy" or "Very Unhealthy" levels. Using integrated observational datasets, the smoke event's spatial extent and temporal persistence provided a unique opportunity to study its atmospheric and surface-level impacts.

Even though satellite observations and chemical transport modeling clearly show the long-range movement of wildfire smoke (Jaffe et al., 2020; Val Martin et al., 2010), much uncertainty remains about the vertical distribution of aged smoke plumes

and how they interact with the dynamics of the regional PBL (National Academy of Sciences, 2022; Ye et al., 2022). Understanding these processes is crucial, especially in the Mid-Atlantic area, where background precursor concentrations and local meteorological variables impact midsummer ozone formation. Previous studies indicate entrainment of the aloft smoke layers. These layers typically transport ozone precursors such as oxides of nitrogen (NO<sub>x</sub>) and volatile organic compounds (VOCs), as well as carbon monoxide (CO), into the diurnally developing PBL; thus, impacting surface ozone concentrations (Dreessen et al., 2016; Lara et al., 2022). To analyze the interaction of the smoke plume with the PBL, it is equally important to distinguish the smoke layer from that of the surface aerosols.

The BWC has been the subject of extensive air quality research given its proximity to NASA Goddard Space Flight Center (GSFC), where reference AERONET (Aerosol Robotic Network) and Pandora instruments are located. Long-term AERONET observations have examined aerosol optical properties and their variability in this region (Eck et al., 2013). Raman lidar systems deployed at GSFC have also provided detailed vertically resolved retrievals of aerosol optical and microphysical properties (Veselovskii et al., 2013) and have been used to characterize transported smoke layers from distant fires such as the 2015 California wildfires (Veselovskii et al., 2015). In addition, the DISCOVER-AQ field campaign provided a comprehensive picture of air-quality dynamics in the BWC, including how meteorological conditions and aerosol hygroscopic growth influence optical and microphysical properties (Pérez-Ramírez et al., 2021). These studies form an important foundation for understanding aerosol-PBL interactions in this region.

Building on this earlier work, lidar-based observations with polarization capability help in providing new opportunities to study the vertical structure and phase composition of aerosol layers, which also act as a proxy for PBL height. Depolarization ratio measurements are used to distinguish wildfire smoke, which is often made up of solid-phase, fractal-like soot aggregates, from liquid-phase boundary layer aerosols and clouds (Haarig et al., 2018; Tesche et al., 2009). Several past studies have utilized automatic lidar ceilometer capabilities to analyze smoke particles (Dreessen et al., 2023; van der Kamp et al., 2008; Wu et al., 2012). However, with the latest addition of depolarization capability in ceilometers, such as the Vaisala CL61, investigations characterizing smoke plumes remain scarce. Our study leverages the Vaisala CL61 ceilometer with depolarization capability to provide continuous, autonomous vertical profiling of wildfire smoke. While multiwavelength Raman lidars yield richer microphysical retrievals, they are resource-intensive and typically limited to specialized research facilities or field campaigns. By contrast, the CL61 offers a more affordable, autonomous solution that can operate continuously, making it particularly well-suited for long-term monitoring and integration into operational air-quality networks. Despite these earlier efforts, there remain some gaps in understanding transported Canadian wildfire smoke over the BWC (Dreessen et al., 2016, 2023; Yang et al., 2022). Even fewer studies have integrated high-resolution ceilometer depolarization data with trajectory analysis to detect smoke plumes, the degree of their mixing, and the impact on surface pollution concentrations. The events of 2015 and 2023 Canadian wildfire smoke making it to BWC and interacting with the PBL underscored two long-standing research needs: (I) a clearer systematic understanding of how elevated smoke layers are transported, entrained, and ultimately mixed into the PBL, and (II) the development of observational strategies capable of resolving the vertical aspect of that process in real-time. In this context, the present study aims to fill a critical observational gap by examining smoke intrusion events over the BWC during the summer of 2023.

This manuscript is structured as follows. Section 2 describes this study's observational and modeling data, comprising the ceilometer-derived backscatter and depolarization profiles, surface air quality measurements, and the HYSPLIT trajectory analysis framework. It also describes the procedure for detecting and classifying smoke intrusion events. Section 3 presents the findings, analyzing four representative case studies for different plume-PBL interaction scenarios. These cases document different degrees of vertical mixing, surface concentration enhancement, and synoptic transport regimes. Section 4 concludes with a synthesis of important findings, implications for air quality prediction in smoke-affected regions, and suggestions for future observational and modeling work for enhancing vertical resolution within wildfire smoke estimation.

## 2 Methodology







## 2.1.1 Study region

This study focuses on the Baltimore-Washington Corridor (BWC), extending approximately 30 miles, which includes the densely populated and industrialized region between Baltimore, Maryland, and Washington, D.C. The BWC has complicated air quality dynamics due to the unique interaction of local urban pollutants, meteorology, and proximity to natural areas such as the Chesapeake Bay (Karle et al., 2024; Dreessen et al., 2016). It is characterized by a dense network of congested highways, railways, industrial facilities as well as the presence of coal-fired power stations (L-W, 2002), all pollution sources which degrade the local air quality through the emission of nitrogen oxides (NO<sub>x</sub>), carbon monoxide (CO), volatile organic compounds (VOCs), and particulate matter (PM2.5); thus, contributing to ground-level ozone and smog formation. The corridor's high population density exacerbates air quality concerns, particularly during events when external pollution sources are transported into the region. Historically, the BWC has experienced episodes of degraded air quality, notably during the 2015 and 2023 Canadian wildfire seasons. This highlighted the region's vulnerability to transboundary air pollution and emphasized the growing impacts of climate change on air quality caused by external occurrences such as smoke transport from wildfires.

## 2.1.2 National Ambient Air Quality Standards (NAAQS) for NO2, PM2.5, O3

Globally, the inhalation of ambient particulate matter (PM) is approximated to cause between 7.5 and 10.3 million premature deaths annually (Manavi, 2025). Fine particulate matter (PM<sub>2.5</sub>) is especially hazardous due to its ability to penetrate deep into lung tissue, deteriorating cellular structures and, with prolonged exposure, contributing to lung cancer, cardiopulmonary mortality, and immune system suppression (Manavi, 2025). The significant health impacts of PM<sub>2.5</sub> underscore the critical need for accurate transportation modeling in the case of distant wildfires, from point sources to densely populated urban areas. The following table shows the current Environmental Protection Agency (EPA) National Ambient Air Quality Standards (NAAQS) that the US needs to comply with for three major air pollutants referenced in this study. Concentrations exceeding

these values harm human health and the environment (EPA, 2024). Exceedances are also violations of the EPA's NAAQS, which are required to be attained by the Clean Air Act.

Table 1: Environmental Protection Agency: NAAQS

| Air Pollutant           | Harmful Concentration Levels              |
|-------------------------|-------------------------------------------|
| Nitrogen Dioxide (NO2)  | 100 ppb averaged over 1 hr                |
| Ozone (O <sub>3</sub> ) | 70 ppb averaged over 8 hrs                |
| PM <sub>2.5</sub>       | 35 μg/m <sup>3</sup> averaged over 24 hrs |

## 2.2 Measurements and Instrumentation



To evaluate the effects of Canadian wildfire smoke on the BWC, we used a multi-instrument observational method that included satellite imagery, ground-based air quality monitoring, wind trajectories, and remote sensing from a ceilometer. The data used in this study were obtained from the Howard University Beltsville Campus (HUBC) (latitude: 39.056, longitude: -76.875). The co-location of ceilometers, radiosonde launches, and surface measurements at HUBC provides a unique dataset for researching aerosol-boundary layer interactions. Detailed descriptions of each measurement approach are provided below.

## 2.2.1 Linear Depolarization and Smoke Detection

- Vaisala CL61 ceilometer is a vertically pointing, single wavelength lidar that measures both attenuated backscatter and volume depolarization ratio (VDR) (also commonly referred to as linear depolarization ratio (LDR)), measuring capability. The LDR is calculated by dividing the perpendicular component by the parallel component from the backscattered signal, without separating aerosol and molecular contributions (Bedoya-Velásquez et al., 2022; Bellini et al., 2024; Inoue & Sato, 2023). The LDR approximates aerosol depolarization in aerosol-dominated layers to distinguish spherical and non-spherical aerosols. This property is widely used in the lidar community to distinguish aerosol types, for example, separating smoke, dust, and cloud particles (Tesche et al., 2009; Haarig et al., 2018).
  - For this study, we extensively used the LDR, one of the key characteristics of the CL61, to identify the smoke plume. However, the LDR is not interpreted in isolation, but in combination with attenuated backscatter (aerosol load), surface air quality, PM<sub>2.5</sub> measurements, satellite imagery, and HYSPLIT trajectories. We used all this information to discriminate between smoke from wildfires and hydrometeors. Days with high aerosol layers detected through ceilometer profiles that showed increased depolarization over the PBL were marked as probable fire smoke occurrences. They were further confirmed with cross-validation by satellite imagery and wind trajectory analysis.
  - To diagnose the vertical distribution and optical nature of the aerosols aloft, we used a case from 5 June 2025 (Figure 1). We paired high-resolution ceilometer measurements with a collocated radiosonde released at 06:32 UTC from the Beltsville site.
- Fifteen-minute averages of the CL61 LDR were compared to the radiosonde's temperature and relative humidity profiles to

verify particle phase. The radiosonde confirmed a deep isothermal layer with maximum humidity between 8-9 km (pink shade), coincident with ceilometer LDR values of  $\sim 0.35$ ; this combination is characteristic of ice crystals and, therefore, attributed to a cirrus cloud deck. In contrast, a relatively dry but above freezing layer between 3-4.5 km aligned (yellow shade) with markedly lower LDR values ( $\sim 0.05$ ), consistent with weakly depolarizing, predominantly spherical biomass-burning smoke particles. Selecting this day when both cloud and smoke layers were present during a routine launch allowed us to cross validate ceilometer-derived phase discrimination against independent thermodynamic soundings and to establish representative LDR thresholds ( $\geq 0.3$  for ice cloud;  $\leq 0.08$  for smoke) used throughout the case study analysis.



Figure 1. Vertically resolved aerosol and thermodynamic observations from HU-Beltsville on 5 June 2025. (**Top left**): Time-height cross section of attenuated backscatter coefficient (β (m sr)<sup>-1</sup>) from the Vaisala CL61 ceilometer, illustrating a persistent smoke layer between 3-4 km and elevated cirrus cloud structures between 8-12 km before 10 UTC. (**Bottom left**): Corresponding LDR, highlighting enhanced depolarization between 2-4 km, indicative of non-spherical aerosol particles

(smoke), and elevated LDR values near 8-9 km, consistent with optically thick, ice-containing cloud layers. The black vertical line at 06:32 UTC indicates a radiosonde was launched from the site. (**Right**) Vertical profiles from the radiosonde: (**column 1**) air temperature (K), (**column 2**) relative humidity (%), (**column 3**) total backscatter, and (**column 4**) LDR averaged from ceilometer data over a 15-minute window centered on the launch time. Blue horizontal dashed lines mark altitudes and shaded regions of the identified smoke layer (yellow) and cloud (pink) layers.

## 2.2.2 Satellite Observations





True-color satellite images (https://worldview.earthdata.nasa.gov) taken by the Visible Infrared Imaging Radiometer Suite (VIIRS) on board NOAA-20 were employed to identify the regions of the fire sources and evaluate the horizontal extent and transport paths of upper-level smoke plumes (NASA, 2025). Satellite images of the North American continent offered visual verification of large-scale smoke cover and helped monitor its transport towards the Mid-Atlantic region. Days identified by the Vaisala CL61 ceilometer as having a high aerosol layer above the PBL were further analyzed with images from VIIRS to confirm the presence of transported smoke. This combined methodology ensured that only those instances with elevated aerosol detection by ceilometer and satellite-verified smoke signatures were chosen for detailed analysis.

# 2.2.3 HYSPLIT Back-Trajectories

NOAA's HYSPLIT (Hybrid Single Particle Lagrangian Integrated Trajectory) model provides backward trajectory analyses, crucial for identifying the potential source regions and transport pathways of air parcels arriving at a specific location. It is extensively used for analyzing wind trajectory, dispersion, and deposition problems (Draxler & Hess, 1998; Stein et al., 2015). We computed 72-hour backward trajectories for the Baltimore-Washington Corridor to diagnose transport pathways during the smoke-affected periods, but focused interpretation on the final 24-48 hr when the trajectory uncertainty is smaller (Stein et al., 2015; Stohl, 1998). Howard University Beltsville Campus (HUBC), the study site (latitude: 39.05 and longitude: -76.87), was the source for these back trajectories. All the trajectories were computed for three arrival heights: 500 m, 1500 m, and 2000 m above ground level (AGL), using Global NCEP GFS 0.25 forecasts (3-hourly), which avoid premature termination at lateral boundaries and provide complete coverage over North America. We interpret back trajectories as pathway hypotheses conditioned on the driving meteorology, not as proof of emission.

#### 2.2.4 Continuous Ambient Air Monitoring Station

Maryland's Department of the Environment (MDE) has several continuous ambient monitoring stations (CAMS) scattered throughout Maryland. One of their supersites, part of the EPA's Chemical Speciation Network (CSN), is hosted within Howard University's Beltsville campus, right along the BWC (Karle et al., 2024). The university collaborates with MDE, monitoring continuous air quality, including particulate matter speciation, PM<sub>2.5</sub>, O<sub>3</sub>, and NO<sub>x</sub> measurements. This site plays a key role in monitoring atmospheric aerosols in the Mid-Atlantic region of the USA. It contributes to understanding regional air quality

and long-range transport of pollutants on public health and our evolving climate. Data collection from this site has been analyzed as part of the backbone of this research.

## **2.2.5 AERONET**




NASA's AERONET (Aerosol Robotic Network) is a globally distributed network of remote sensing instruments, such as ground-based sun photometers, used to study the optical properties of aerosols in the atmosphere (AERONET, 2020). One of the most important aerosol features studied is the Aerosol Optical Depth (AOD). This measurement sheds light on the concentration and nature of aerosols in a particular region of the sky. AERONET open-access data allows in situ observations of aerosol optical, microphysical, and radiative properties and characterization of atmospheric particulate composition. It also permits scientists to validate satellite data retrievals, improve climate models, and quantify aerosol radiative forcing on the climate system (AERONET, 2020). The primary purpose of employing AERONET data in this analysis was to identify and characterize smoke events that occurred during the transport of the smoke from the Canadian wildfire. We analyze AERONET Version 3 data, specifically at Level 2.0 (quality-assured). We used the Ångström exponent, calculated from the AOD at different wavelengths (often including 440 nm and 870 nm). It offered an indication of the dominant size of aerosol particles; higher values (typically > 1.5) suggest a greater proportion of fine particles, characteristic of smoke (Schuster et al., 2006).

#### 215 2.2.6 Canadian Fire Source Data

We obtained approximate fire source locations from the Canadian Interagency Forest Fire Center (CIFFC) via the Canadian Wildland Fire Information System (CWFIS) (https://cwfis.cfs.nrcan.gc.ca/). These maps and data products are based on the best available fire reports but may not always reflect the current fire situation in real time. To complement these maps, we also examined the VIIRS Fire and Thermal Anomalies product (Day, 375 m) from the NOAA-20 satellite, which provides near-real-time fire detections and thermal anomaly hotspots. Supplementary Figures S1–S4 show the VIIRS fire detections and the CWFIS fire maps for each case day, with the highlighted HU-Beltsville site. This study employed CIFFC fire locations, HYSPLIT trajectories, synoptic weather maps, and satellite images to find the places that best matched the observed smoke plumes and their transport patterns. This multi-evidence method avoids assigning " origin " to trajectory analysis.

#### 3 Results

The devastating 2023 Canadian wildfires and widespread air quality impacts across the eastern United States were extraordinary (Zhang et al., 2025). Multiple smoke intrusions were observed at HU-Beltsville from May to July. However, the four episodes discussed below were selected for detailed analysis because they capture the contrasting regimes of smoke plume-PBL interaction observed during that period. These cases are not exhaustive but represent the dominant transport scenarios affecting the BWC during that summer.

## 230 3.1 Case 1: 22-26 May 2023




At the end of May 2023, the wildfires in western Canada were brewing, and the smoke spread long distances. The satellite image from NOAA-20 provided a comprehensive overview of the wildfire smoke transport from Canada into the mid-Atlantic region, as seen in Figure 2(a). A notable greyish-white smoke extended across much of western and central Canada and plunged into the northern USA. The red-outlined region in the upper-left portion of the image highlights the core area of active wildfires in May 2023 based on the information from the Canadian Wildland Fire Information System (https://cwfis.cfs.nrcan.gc.ca/). The plume appears to flow eastward, following upper-level atmospheric circulation. It expanded into a broad haze covering large sections of the Canadian prairie provinces before stretching into the USA Midwest. NASA's Modern-Era Retrospective Analysis for Research and Applications version 2 (MERRA-2) revealed a predominantly westerly flow over North America. Figure 2 (b) illustrates the atmospheric dynamics at the 500 hPa level from 21-24 May 2023, demonstrating the evolving synoptic-scale conditions that facilitated the transport of wildfire smoke. On May 21, a pronounced trough over the northwestern USA initiated a strong north-westerly wind along its axis, effectively channeling smoke southward. Over the next few days, the trough deepened and expanded eastward. The jet stream also facilitated the long-distance transmission of dense smoke. Concurrently, a downstream ridge in the eastern USA created steady subsidence conditions, allowing the smoke to settle and concentrate over the mid-Atlantic region. This allowed for the development of optimal synoptic conditions for wildfire smoke propagation over large distances. As a result of the arrival of the smoke plumes, regional air quality was projected to deteriorate, particularly in the BWC region.

Figure 2. (a) Satellite imagery from NOAA 20/VIIRS with corrected reflectance for 21 May 2023, with the red dashed rectangle showing the region with active fires northwest of Canada (VIIRS Characterization Support Team, 2016). (b) Wind speed and directions at 500 hPa from MERRA-2 from 21-24 May 2023.

The HYSPLIT backward trajectory analysis for 1200 UTC on 25 May 2023 (Figure 3) demonstrated that air masses arriving in the BWC region at multiple altitudes (0.5 - 4 km) originated from regions impacted by Canadian wildfire smoke, as seen in (Figure 2a). Most trajectories show a direct, coherent south-eastward pathway from Quebec into the mid-Atlantic, indicating efficient boundary layer transport of smoke-laden air. Some elevated trajectories revealed a more complex pattern, with some parcels descending from higher altitudes while others passing over the Great Lakes before reaching the study site. Thus, highlighting the role of subsiding elevated smoke layers. Few trajectories, originating west over the upper Midwest, also converge towards the BWC region, albeit remaining generally above the surface boundary layer throughout their transport. Overall, the trajectory ensemble confirms that the study region sat downstream of multiple smoke-laden



airstreams, with the vertical distribution of those plumes largely controlled by synoptic subsidence and boundary-layer entrainment.

Figure 3. HYSPLIT 72-hour back trajectories ensemble for air parcels arriving at the study site between 23 May 00 UTC and 25 May 1200 UTC.


While HYSPLIT traced the horizontal and vertical origin of the smoke, it provided limited information on the plume's subsequent interaction with the evolving PBL. This aspect was resolved by analyzing the CL61 measurements (Figure 4), whose attenuated backscatter profiles and LDR documented the temporal apparent lowering over the study site of the aerosol layer. On 21 May, a minimal aerosol backscatter signal, characterized by low depolarization ratios, was observed within and

above the PBL, indicative of relatively clear conditions. However, as time progressed from 22 May onwards, as indicated in the same figure. The backscatter data reveal the emergence of a distinct aerosol layer aloft, approximately between 3 and 6 km altitude, exhibiting depolarization ratios between 0.05 and 0.1. This elevated depolarization signal is consistent with non-spherical smoke particles, corroborating satellite observations that showed extensive smoke transport from Canadian wildfires towards the mid-Atlantic region.








By May 23 and 24, the smoke plume appeared lower over the site and penetrated the PBL, clearly visualized as a layer below 2 km altitude around 24 May 21 UTC – 25 May 03 UTC. The pronounced increase in depolarization ratio within the boundary layer during these periods (ratios reaching up to ~ 0.1) suggests significant intrusion and mixing of smoke aerosols into the lower troposphere, coinciding with a reduction in the PBL turbulence and mixing after sunset as the atmosphere stabilizes. Notably, around 03 UTC on 25 May, a strong aerosol signal is observed, corresponding precisely to a frontal passage confirmed by synoptic maps. Interestingly, despite the temporal coincidence between the smoke intrusion into the boundary layer and the arrival of this cold front, surface observations did not show a significant spike in PM concentrations (Figure 4). This absence of surface PM level observed at the study site is most likely caused by the cold front effectively flushing the smoke away. The 21-26 May 2023 case revealed intricate connections between meteorological dynamics and aerosol dispersion. Such situations show the complex link between synoptic events and aerosol behavior, underlining the significance of combining observational methods for comprehensive atmospheric characterization.

Based on ceilometer evidence of smoke apparent lowering into residual and mixed layers, we assessed the impact of these aloft smoke plumes on surface air quality. Figure 4 presents a time series of ozone (O<sub>3</sub>), nitrogen dioxide (NO<sub>2</sub>), and particulate matter PM<sub>2.5</sub> concentration measured at the study site from 21-27 May 2023, overlaid with wind vectors, Aerosol Optical Depth (AOD), and Angstrom Exponent (AE). The separation of the 440 and 870 nm AOD values and elevated AE values suggests a predominance of fine-mode aerosols, consistent with smoke presence. The shaded portion indicates the smokeintrusion window identified using the ceilometer backscatter (24 May 21 UTC – 25 May 06 UTC), while the pink dashed line marks the cold-front passage at around 25 May 03 UTC. Prior to the arrival of the smoke plume, diurnal photochemistry dominated surface trace-gas behavior, as seen from the in-situ observations (Figure 4). Ozone built up each afternoon to 45-55 ppb under strong isolation, while NO<sub>2</sub> remained below 5 ppb except for a brief pulse on 22 and 24 May, before the plume's arrival. PM<sub>2.5</sub> data was unavailable until late 23 May and hovered below 20 μg m<sup>-3</sup>, indicative of a clean regional background. When the ceilometer showed the apparent lowering of the smoke layer intersecting the residual layer, the PM<sub>2.5</sub> level rose to 12-15 µg m<sup>-3</sup>— three to four times the baseline, yet still well below NAAQS limits (Table 1). The increase coincided with very light, variable winds (< 0.3 m s<sup>-1</sup>), a condition favorable for vertical mixing of lofted smoke into the stagnant surface layer. NO<sub>2</sub> responded modestly (peaking near 6 ppb), whereas O<sub>3</sub> dipped to < 10 ppb overnight owing to titration under shallow, stable conditions. Immediately following the frontal passage, wind speed jumped to  $\sim 1 \text{ m s}^{-1}$  and veered to a persistent northerly. Within two hours, PM<sub>2.5</sub> dropped below 5 µg m<sup>-3</sup> despite continued ceilometer evidence of smoke aloft, confirming that the front effectively flushed the near-surface layer. Daytime O<sub>3</sub> recovered to > 40 ppb under cleaner post-frontal air, demonstrating that regional photochemistry quickly re-established once the smoke was displaced.

Figure 4. Top: Ceilometer LDR (21-27 May 2023 UTC) showing a wildfire smoke layer apparent lowering into the PBL.

Middle: Surface O<sub>3</sub>, NO<sub>2</sub>, and PM<sub>2.5</sub> time series with wind vectors, highlighting smoke intrusion on 25 May. Corresponding attenuated backscatter profiles representing aerosol load are provided in the Supplement (Fig. S5). LDR is shown here as a discriminator to help separate smoke from boundary-layer aerosols and clouds. Bottom: AOD at 440 nm and 870 nm and Ångström exponent, suggesting enhanced fine-mode aerosol loading during the smoke event.

## 315 3.2 Case 2: 05-10 June 2023

Having established the processes that governed Case 1 in late May, we now turned to the early June episode, which differed in both synoptic setting and surface impact. These synoptic meteorological patterns created optimal conditions for extensive smoke dispersion. Satellite image on 6 June 2023 (Figure 5a) revealed a broad whitish-grey haze stretching from southern Quebec across the Great Lakes into the mid-Atlantic. The diffuse signature and areal extent point to well-mixed smoke that had traversed from an intense fire region in eastern Canada (red dashed rectangle) in Figure 5a (left). The MERRA-2 500 hPa analysis revealed a critical synoptic configuration characterized by a persistent high-pressure ridge over the central USA and a strong low-pressure system over the Canadian western coast. This pressure pattern established a pronounced north-westerly flow, creating an efficient transport corridor from the Quebec fire regions into the Great Lakes and subsequently toward the BWC.

During June 5–8, 2023, a stationary low-pressure system off Canada's east coast maintained cyclonic circulation over the region. This circulation transported smoke-laden air masses southward along the system's western periphery. At the same time, large-scale subsidence associated with the downstream ridge promoted the downward mixing of elevated smoke layers, coupling them with the boundary layer and thereby increasing the potential for surface-level air quality impacts. The HYSPLIT back trajectories terminating at 2100 UTC on 6 June 2023 (Figure 5a (right)) confirm the impact of the synoptic forcing; the descending motion associated with high-pressure systems compresses and warms air masses, enhancing boundary layer entrainment processes. In contrast to the May event, this case's trajectories display a stronger westerly component, with air parcels sweeping across Lake Superior and the Lower Peninsula of Michigan before entering the mid-Atlantic region. The lowest trajectory spent extended time (

(a)


(b)

**Figure 5. (a)** Left: Satellite image on 06 June 2023 showing the wildfire smoke plume and fire source (red dashed box) (VIIRS Characterization Support Team, 2016). **Right:** 72-hour HYSPLIT back trajectories ending at 2100 UTC 06 June 2023. (b) Wind speed and directions at 500 hPa from MERRA-2 from 05-08 June 2023.





The depolarization ratios for 5-10 June (Figure 6) show a remarkably coherent slab that originated near 4 km at 00 UTC on 6 June and appeared to lower almost linearly, crossing 2 km at  $\sim 1000$  UTC before reaching < 1 km by 1500 UTC. The implied subsidence rate mirrors the downward motion identified by the HYSPLIT trajectories in Figure 3. As soon as the layer intersected the top of the mixed layer, backscatter intensities below 500 m strengthened sharply, indicating that the elevated aerosol was rapidly entrained. From 9 June onward, the ceilometer recorded a separate, spectrally distinct return between 3 and 5 km, which was identified as the mid-level clouds rather than a smoke plume.

Fine particulate concentrations begin to climb after 12 UTC on 6 June, coinciding with the ceilometer's first detection of the plume (shaded) penetrating the PBL (Figure 6). While the ceilometer does not provide particle size information, the simultaneous rise in PM<sub>2.5</sub> concentrations supports the interpretation that the entrained layer contained fine-mode smoke particles. During this interval, surface winds were exceptionally weak (< 0.2 m s<sup>-1</sup>) and variable, suppressing horizontal dilution and allowing local accumulation. Ozone remained anomalously low (< 15 ppb) throughout the smoke episode, consistent with both attenuated photolysis under the dense plume and titration by co-transported NO<sub>x</sub> (NO<sub>2</sub> peaks  $\approx$  15 ppb). PM<sub>2.5</sub> climbed steadily to  $\sim$  40 µg m<sup>-3</sup> by late 6 June in tandem with light ( $\leq$  0.6 m s<sup>-1</sup>) north-westerly winds, affirming local accumulation of the mixed-down plume.

A much larger PM2.5 level spiked ( $\sim 215~\mu g~m^{-3}$ ) on 8 June, followed by the return of stagnant conditions, but occurred well after the primary apparent lowering of the plume. The increase coincided with the ceilometer's cloud-top signature at 3-5 km (Figure 6). While the cloud layer was not a particulate source, its presence may have reduced surface heating and modified humidity levels, potentially influencing boundary-layer stability and inhibiting dispersion. However, our data shows no clear, direct PBL response to the clouds. Disentangling the relative roles of continued smoke advection versus in-situ processes will require chemical speciation and additional trajectory analysis beyond the scope of the present study.

**Figure 6. Top:** Ceilometer LDR (5-10 June 2023 UTC) showing a wildfire smoke layer apparent lowering into the free troposphere early on 5 June and later cloud cover around 9-10 June. Corresponding attenuated backscatter profiles representing aerosol load are provided in the Supplement (Fig. S6). LDR is shown here as a discriminator to help separate smoke from boundary-layer aerosols and clouds. **Middle:** Surface O<sub>3</sub> (blue), NO<sub>2</sub> (green), and PM<sub>2.5</sub> (red) time series with wind vectors, highlighting the smoke-intrusion period (shaded) on 5–6 June. **Bottom:** AOD at 440 nm (orange) and 870 nm (brown) with Ångström exponent (purple), indicating elevated fine-mode aerosol loading during the smoke event.

## 3.3 Case 3: 15-19 June 2023




Case 3 captured a multilayer smoke incursion into the BWC that resulted from two geographically distinct Canadian fire complexes (West and Central) and differential transport pathways aloft and within the PBL (Figure 7a (left)). The NOAA-20 true-color imagery from 15 June 2023 shows dense plumes emanating predominantly from the western and central parts of Canada (red dashed boxes) (Figure 7a (left). The synoptic system observed during case 3 (15-19 June 2023) was complicated than cases 1 and 2. The high-pressure ridge shifted north-eastward of Canada, while a deep low-pressure system eventually developed over the north-eastern USA. This system maintained cyclonic flow that transported smoke southward, drawing contributions from both fire regions (west and east of Canada) but along different pathways (Figure 7b).

HYSPLIT back trajectories ending at 0000 UTC on 16 June 2023 illustrate this structure (Figure 7a (right)). Trajectories linked to the western fires were carried mainly in elevated layers, guided by anticyclonic flow around the upstream ridge. In contrast, trajectories from central Canada fed into the lower levels through direct southward transport. Differences in vertical motion are also evident. The mid-level back trajectories descended gradually, consistent with large-scale subsidence behind the weakening ridge. However, the near-surface trajectory dropped sharply during the final 12 hours, likely influenced by mesoscale subsidence tied to a shortwave trough or surface high. The trajectory altitude profiles (bottom panel) show these layers remained separated through most of the transport but converged rapidly near arrival. This points to a breakdown of stratification and stronger vertical mixing, probably from boundary-layer deepening driven by surface heating and mechanical turbulence.

(a)

Figure 7. (a) Left: True color satellite image on 15 June 2023 UTC showing the wildfire smoke plume over the mid-Atlantic and three prominent sources of fire (red dashed square) (VIIRS Characterization Support Team, 2016). Right: 72-hour HYSPLIT backward trajectories ending at 00 UTC 16 June 2023 for air parcels. (b) Wind speed and directions at 500 hPa from MERRA-2 from 15-18 June 2023.



On 15 June, around 1800 UTC, the LDR profiles revealed three distinct smoke layers between 4 and 7 km altitude. As the labeled markers in Figure 8 indicate, the elevated smoke layer demonstrated an apparent lowering toward lower altitudes, consistent with trajectory evidence of gradual subsidence. The depolarization values during this evident lowering of the plume ranged between 0.05 and 0.10. Once again, these values provided significant information and confirmed that the depolarization measurements can effectively differentiate between different types of aerosols based on particle shape and composition, especially in the free atmosphere (CL61 White Paper, 2021). Between 06-13 UTC of 16 June, the smoke interacted with the residual PBL, as seen based on the depolarization signals at the interface. Precipitation was also recorded around 15 UTC on 16 June, as indicated by the higher depolarization ratio values (0.15-0.25). We do not observe a significant smoke signal during that time. The smoke plume is seen again between 2-3 km and 1900-2300 UTC on 16 June. The smoke seemed present in the residual layer and later interacted with the emerging convectively driven PBL on 17 June. In the figure, PBL is also visible as the region with varying depolarization values and reaching an average height of 2-2.5 km during this case study.

Surface ozone levels gradually declined from 65 ppb to 10 ppb during the smoke intrusion phase, with a gradual increase in the NO<sub>2</sub> levels. The ozone levels recovered on 16 June at 0900 UTC by going up 5 ppb. However, they dropped again, likely due to the NO<sub>x</sub>-mediated titration, as the smoke plume is known to carry substantial NO<sub>x</sub> emissions, as indicated by increasing levels of NO<sub>2</sub>. Simultaneously, we noticed increased PM<sub>2.5</sub> levels, especially after the smoke intrusion into the PBL. The

- increasing ozone levels after 1200 UTC of 16 June slowed down between 1200 and 1500 UTC. They could be attributed to aerosol-induced UV attenuation and some cloud coverage suppressing photochemical ozone production. Ozone level recovered after the warm front passage (denoted by pink dashed line) and was associated with a wind shift, which may have introduced an urban NO<sub>x</sub>/VOC mixture that synergized with smoke-derived precursors to reignite photochemical ozone formation. During the pre-frontal period, weak winds reduced horizontal transport, allowing the descending plume to remain over the site and mix into the boundary layer. Post-frontal winds promoted horizontal advection of the smoke. As observed near the surface, the wind shifts aligned with the warm front's passage, which is followed by brief precipitation, as seen by the strong strip of LDR (green, yellow, and red). This likely caused a brief collapse of the PBL, which rapidly recovered around 1700 UTC. By 1800 UTC, the evolving convectively driven PBL was coupled with the residual layer comprising a smoke plume. The AOD values at 440 and 870 nm were relatively higher throughout 16 June, confirming the dominance of smoke aerosols in the lower atmosphere.
- In this case, ozone exceeded 65 ppb on 15 June before the intrusion of the smoke plume but then dropped to 10 ppb as the plume intersected the PBL overnight. On 16 June, ozone recovery was uncommon, with two small step-like plateaus instead of smooth growth as seen on other surrounding days. This suggests that photochemical synthesis was partially inhibited by smoke in the system. Elevated NO<sub>2</sub> levels during the same period indicate titration effects, which would have further limited ozone growth. However, following the frontal passage, ozone gradually rebounded, while PM<sub>2.5</sub> steadily increased, reaching ~20-25 μg m<sup>-3</sup> by 1800 UTC on 17 June. Although smoke layers lingered into 18 June, stronger surface winds kept O<sub>3</sub> maxima capped near 50 ppb.

Figure 8. Top: Ceilometer LDR (15-19 June 2023 UTC) showing wildfire smoke layer apparent lowering into the PBL, followed by cloud and precipitation on 16 June, and a separate elevated smoke plume on 17 June. Corresponding attenuated backscatter profiles representing aerosol load are provided in the Supplement (Fig. S7). LDR is shown here as a discriminator to help separate smoke from boundary-layer aerosols and clouds. Middle: Surface O<sub>3</sub> (blue, left axis), NO<sub>2</sub> (green, left axis), and PM<sub>2.5</sub> (red, right axis) with wind vectors, highlighting the 15–16 June smoke intrusion (shaded pink) and a warm front passage (dashed magenta line) on 16 June. Bottom: AOD at 440 nm (orange) and 870 nm (brown) with Ångström exponent (purple, right axis), indicating enhanced fine-mode aerosol loading during the elevated smoke episodes.

#### 3.4 Case 4: 28 June to 1 July 2023

Satellite true-color imagery on 28 June (Figure 9a (left)) depicts a dense smoke plume from active fire sources in eastern Canada extending southeastward into the mid-Atlantic. This distribution reflected the combined influence of a deep low-pressure trough over the north of Hudson Bay region and a strong high-pressure ridge anchored over the central USA. The resulting 500 hPa flow produced a meridional transport pattern, with elevated smoke carried southward along the trough's western flank and lower-level plume components influenced by the subsiding branch of the ridge (Figure 9b).

HYSPLIT back trajectories ending at 1200 UTC on 28 June are consistent with this synoptic setup (Figure 9a (right)). Parcels initialized at 3–3.5 km gradually descended under ridge-related subsidence and only approached the boundary layer during the final 12 hours. In contrast, mid-level parcels descended more sharply ahead of the ridge axis and spent extended time within the Ohio Valley boundary layer. This trajectory behavior indicates that lower-level air masses were exposed to conditions favorable for vertical coupling well before reaching the BWC. The altitude-time profiles highlight this distinction: elevated smoke layers remained separated aloft until close to arrival, while the lowest trajectories underwent mixing. It is highly likely that part of the smoke became well-mixed into the PBL upstream and was subsequently advected downstream toward the study site. Consistent with this pathway, the CL61 detected no elevated LDR signature in the 3-6 km layer, confirming that the smoke was already well mixed into the regional PBL upon arrival.

(a)

**Figure 9. Left:** True color satellite image on 28 June 2023 UTC showing the wildfire smoke plume over the mid-Atlantic and one of the prominent sources of fire (red dashed square) (VIIRS Characterization Support Team, 2016). **Right:** 72-hour HYSPLIT backward trajectories ending at 1200 UTC 28 June 2023 for air parcels.

Surface observations respond promptly to this low-level influx. Beginning near 0900 UTC on 28 June, PM<sub>2.5</sub> rose steadily, surpassing 60  $\mu$ g m<sup>-3</sup> by late evening, while ozone peaked at ~ 55 ppb in the afternoon before collapsing below 10 ppb overnight (Figure 10). These divergent trends suggest that the incoming pollutants carried a substantial primary PM load but only moderate ozone precursors; nocturnal titration by accumulated NO and reduced photolysis likely contributed to the sharp ozone decline (Asmar et al., 2025). The following day, a weaker north-westerly flow sustained the PM built up. The PM<sub>2.5</sub> levels exceeded 130  $\mu$ g m<sup>-3</sup> by 1800 UTC on 29 June, whereas ozone exhibited a two-step rise, first to ~ 80 ppb, then, after a brief lull, to > 90 ppb. The midday ozone dip coincided with a transient wind speed and direction.

Surface wind-vector analysis provided the dynamical context for the increasing levels of pollutants. On 28 June, moderate north-westerlies (0.4 - 0.8 m/s) channeled the smoke directly into the BWC. Night-time winds slackened to < 0.2 m/s, promoting pollutant stagnation and an observed plateau in PM<sub>2.5</sub> levels. Around 1500 UTC on 29 June, the surface flow veered abruptly to southerly; within three hours, PM<sub>2.5</sub> levels began to fall, confirming the ventilation by cleaner maritime air. By 30 June, persistent southerlies exceeded 1m/s and both PM<sub>2.5</sub> and ozone levels dropped to values well below the previous day's maxima. Compared with the earlier May and early June cases, these cases illustrated a distinct mechanistic pathway: smoke advected entirely within the mixed layer can yield extreme particulate concentrations without an accompanying elevated layer signature in lidar, and its surface impact is modulated primarily by horizontal wind shifts rather than the vertical mixing process.

Figure 10. Top: Ceilometer attenuated backscatter (28 May 00:00 UTC–2 June 02:00 UTC 2023) displaying persistent cloud layers between ~1–3 km and intermittent elevated returns. Corresponding attenuated backscatter profiles representing aerosol load are provided in the Supplement (Fig. S8). LDR is shown here as a discriminator to help separate smoke from boundary-layer aerosols and clouds. Middle: Surface O<sub>3</sub> (blue, left axis), NO<sub>2</sub> (green, left axis), and PM<sub>2.5</sub> (red, right axis) time series with overlaid wind vectors. Bottom: AOD at 440 nm (orange) and 870 nm (brown) with Ångström exponent (purple, right axis), indicating variations in fine-mode aerosol loading over the same period.

#### 4 Discussion and Conclusions

The 2023 Canadian fire season delivered three mechanistically distinct smoke episodes to the Baltimore–Washington Corridor: (i) lofted-subsiding transport (24–25 May and 15-19 June), (ii) subsidence-entrainment under stagnation (6 June), and (iii) advection-dominated boundary-layer transport (28 June-1 July). Each case unfolded under a unique combination of synoptic forcing, boundary-layer depth, and local wind regime, producing markedly different pollutant footprints at the surface. In the May case, an elevated plume interacted transiently with the nocturnal residual layer but was rapidly flushed by a post-frontal northerly surge; PM<sub>2.5</sub> never exceeded 15 µg m<sup>-3</sup>, and ozone rebounded once cleaner air arrived. The 6 June 505 episode highlighted the importance of timing: a smoke slab apparently lowered into a shallow, weak-wind PBL, allowing  $PM_{2.5}$  to exceed 150 µg m<sup>-3</sup> while photolysis suppression kept  $O_3$  anomalously low. The late June event demonstrated that extreme particulate loading could arise even without an elevated lidar signature when smoke is advected entirely within the mixed layer; here, horizontal wind veers, rather than vertical mixing, controlled the pollution trajectory. These contrasts 510 emphasize that public health risk cannot be inferred solely from satellite optical-depth fields or an elevated aerosol layer detected by lidar. The vertical stage at which the plume arrives, and the contemporary ventilation state of the boundary layer, jointly determine whether smoke aloft remains a distant spectacle or becomes an acute surface hazard. A key contribution of this study is the first systematic use of the Vaisala CL61 depolarisation channel over the Mid-Atlantic to disentangle wildfire smoke from other aerosol types. While ceilometer backscatter alone can misclassify dense 515 boundary-layer haze or low clouds as smoke, the LDR of biomass-burning particles ( $\approx 0.05-0.15$ ) provides a robust discriminator. In the May and early June cases, elevated LDR signatures were pivotal in confirming the apparent lowering over the study site of non-spherical smoke into the PBL, a distinction that conventional backscatter or satellite AOD could not securely provide. Conversely, the absence of an LDR enhancement during the late June event corroborated the trajectory analysis, indicating that the plume was already embedded in the mixed layer. These findings highlight the potential value of 520 including depolarization-capable, affordable commercial lidars in regional air quality networks and suggest that near-realtime LDR fields could serve as a complementary constraint for smoke forecast systems. The four episodes illustrate how surface air quality responses depend less on the smoke aloft than on the timing of its interaction with the PBL. When smoke was embedded in the daytime mixed layer (case 4), surface PM2.5 levels rose to >100 ug m<sup>-3</sup> and ozone reached a critical level of > 90 ppb. When an elevated layer intersected the shallow nocturnal or early 525 morning PBL (case 2), PM<sub>2.5</sub> exceeded 150 μg m<sup>-3</sup>, but ozone remained suppressed by reduced photolysis and titration. In cases 1 and 3, we made contrasting observations, where lofted smoke plumes briefly interacted with the residual or growing mixed layer, producing modest and short-term PM<sub>2.5</sub> enhancements ( $< 20 \,\mu g \, m^{-3}$ ) with ozone responses largely governed by frontal dynamics. Across all cases, the interplay of transport regime, PBL depth, and ventilation state determined whether the smoke intrusions remained aloft or translated into acute surface exposure. Across the four cases, Canadian wildfire smoke 530 reached the BWC through different transport regimes, including direct boundary-layer advection, subsidence-driven

lowering of lofted plumes, and multilayer convergence from geographically separated fire complexes. A recurring feature, however, was the importance of when and where the plume intersected the PBL, whether locally over the BWC or upstream. For operational air quality models not to over-predict ozone when large smoke burdens coincide with high photolysis suppression or under-predict PM<sub>2.5</sub> when smoke layers appear to lower abruptly overnight, the case studies here suggest possible avenues for improvement. The height-resolved emission placement accurately representing pyro-convection injection heights is essential to reproduce the subsequent subsidence timelines observed in the May and June events. Capturing the diurnal growth—collapse cycle and its interaction with weak-wind stagnation, as evidenced on 6 June. Assimilating ceilometer-derived PBL height and LDR can constrain model dilution rates and aerosol speciation, improving PM<sub>2.5</sub> forecasts during advection-dominated episodes. We also want to emphasize that this analysis is limited to a single lidar site and relies on total PM<sub>2.5</sub> concentrations without detailed elemental carbon or VOC tracers. Multi-lidar networks coupled with airborne in-situ sampling would permit full three-dimensional mapping of smoke layers and their chemical aging. Additionally, direct measurements of photolysis rates would clarify the relative contributions of smoke shading versus precursor availability to the observed ozone responses.







Three transport regimes—subsiding lofted layers, stagnation-aided entrainment, and boundary-layer advection—produced markedly different surface impacts despite similar satellite optical signatures.  $PM_{2.5}$  ranged from  $< 15 \mu g m^{-3}$  in the ventilated May case to > 130 μg m<sup>-3</sup> under late-June stagnation, while O<sub>3</sub> effects varied from titration-driven minima (< 15 ppb) to photochemically driven maxima (> 90 ppb). Depolarization lidar provided a useful discriminator for distinguishing lofted smoke layers from boundary-layer aerosols and clouds, but detailed optical and microphysical characterization requires multiwavelength Raman or HSRL observations. LDR thresholds allowed discrimination between elevated smoke, boundary-layer aerosols, and cloud tops, reducing false positives in smoke detection. The timing of plume-PBL intersection relative to local wind and stability conditions emerged as the dominant control on surface exposure. Forecast guidance should prioritize accurate simulation of PBL dynamics and incorporate near-real-time lidar constraints. Taken together, these cases suggest that surface air quality impacts in the mid-Atlantic depend less on the simple presence of smoke aloft than on the combination of synoptic flow, entrainment pathways, and PBL conditions along the transport route. Distinguishing whether smoke has already mixed into the boundary layer upstream or remains elevated until arrival is therefore important for anticipating when transboundary smoke will influence surface concentrations. Satellite AOD alone is insufficient to gauge surface risk. Remote-sensing products must be blended with height-resolved observations and trajectory analyses to resolve whether smoke is aloft, entraining, or already well mixed. Future research should combine multi-site depolarization lidar, chemical speciation, and data-assimilative modeling to refine exposure estimates and to develop targeted health advisories during transboundary smoke events. In an era of intensifying boreal wildfire activity, such interdisciplinary approaches are essential for safeguarding air quality in the densely populated mid-Atlantic and beyond.

## **Data Availability**

The datasets and visualization materials used in this study are available from the corresponding author upon reasonable request.

The authors plan to make the data publicly accessible through a recognized repository.

## **Author Contributions**

NNK, RKS, and SC performed the conceptualization. RR and AF provided the ground measurements, and NNK and RKS curated the data. NNK and RR completed the methodology. NNK and RKS performed the thorough investigation. NNK, RKS, and SC performed the analysis. NNK drafted the original paper, which was reviewed by RKS and SC and edited by RR. The entire team contributed significantly to improving this work. SC and RKS acquired funding for this project.

#### **Competing Interests**

The contact author has declared that none of the authors has competing interests.

#### Acknowledgements



The authors are thankful to the research staff at Howard University Beltsville Campus (HUBC), Maryland Department of the Environment (MDE), and Environmental Protection Agency (EPA) for supporting the data collection and archiving. The authors thank the NOAA Cooperative Science Center for Atmospheric Sciences and Meteorology (NCAS-M) for continuous institutional support.

## **Financial Statement**

This research has been supported by the U.S. Department of Commerce, National Oceanic and Atmospheric Administration (NOAA), through the Educational Partnership Program (grant no. NA22SEC4810015) and the NOAA Cooperative Science Center for Atmospheric Sciences and Meteorology (NCAS-M). Additional support was provided by the National Aeronautics and Space Administration (NASA) under the MUREP/DEAP program (grant no. 80NSSC23M0049) and NASA grant no. 80NSSC22K1405, as well as by the National Science Foundation (NSF) through the HBCU-Excellence in Research program (award no. 2000201).

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
