# Peer review of "Vertical Profiling of Canadian Wildfire Smoke in the Baltimore-Washington Corridor - Interactions with the Planetary Boundary Layer and Impact on Surface Air Quality"

_EGUsphere, 2025_

## Author Comment (AC2)

Response to Reviewer CC1

Reviewer comments in black and response in red.

This is a very interesting data set in Beltsville, MD, USA. It appears to be a valuable, specialized resource for studies like this. It will be great to learn more about it and for the science community to have eventual access to these data. Much of the paper deals with "descent" of smoke layers as manifested in lidar time-series data.

» The authors would like to thank the reviewer for the careful reading of our manuscript, constructive positive feedback, and recognition of the value of the Howard University Beltsville Campus (HUBC) dataset. We are happy to provide the data, additional figures, and processed products upon request. We are actively trying to build a public data repository that will make the HUBC dataset broadly accessible to the scientific community. Our webpage contains more information and updates on our data activities: https://hu-bc.org/hubc/

In section 2.2 (Measurements and Instrumentation), we have added a line at the beginning to highlight the novelty of the HUBC measurement site. **Line # 154-156**

I would caution that sloping aerosol and cloud features in such data representations may not be attributable to meteorological forces or sedimentation. A lidar time series simply shows what is blowing overhead at different times. There's no way to know a particle's vertical history or future from such a rolling snapshot of a particulate layer. Please see a comment posted to an ACP paper published back in 2010: https://acp.copernicus.org/articles/10/11921/2010/acp-10-11921-2010-discussion.html

» Thank you for sharing the ACP discussion, it was very helpful. On the "descent" of smoke plume, we fully agree with reviewer's caution. A downward sloping feature in a Eulerian time-height curtain is not proof of subsidence or gravitational settling. As reviewer pointed out in the ACP discussion of a similar case, sloping structure can result from shear/tilt and evolving horizontal gradients blowing through the beam, without additional context a vertical aerosol backscatter curtain alone cannot prove vertical motion. To address this, we have softened our terminology from "descent" to "apparent lowering over the site", unless multiple lines of evidence support subsidence, and we have added that nuance explicitly in section 3 of the revised text.

For some reason, all the HYSPLIT trajectories are shorter than the specified 72 hours. Note the time series below each one. I suspect that this is an artifact of the HRRR data choice. The HRRR data are not global. It can be seen that some of the trajectories end up at about the same latitude in Canada. Maybe that's the edge of the HRRR data grid? I tested one of the scenarios, using GFS global data, and the results came out as 72 hours long. Regardless of the reason, the 72-hr premise is not borne out in the data.

» Good catch. The reviewer rightly pointed out that the HYSPLIT trajectories are shorter than 72 hr. In methods we wrote that all back trajectories were computed for 72 hr. using HRRR (3-km) meteorology (CONUS + southern Canada coverage), but the figures you rightly flagged end before 72 hr. because of HRRR's limited domain causes back trajectories to terminate once parcels leave the grid. This was our oversight; we should have noted the domain limit and used a global field for the plotted cases. We have now rerun all displayed trajectories with GFS to full 72 hr. and updated the captions to list the meteorological driver and duration. This correction does not change our interpretation; we treat trajectories as transport pathways rather than definitive proof of source.

We retain 72 hr trajectories (now rerun with GFS) to preserve synoptic context to Canadian smoke affected regions, but we emphasize the final 24-48 hr in our interpretation and note the associated trajectory uncertainties (Stein et al., 2015; Stohl, 1998). In the revised manuscript, we have corrected Section 2.2.3 and updated Figures 3,5,7, and 9 and their captions accordingly.

Speaking of trajectories, it might not be the case that they show an "origin" any more than a possible path through smoke. There is no information inherent in the trajectories indicating a polluting origin point. The fact that the trajectories illustrated in the paper are all shorter than stated adds to the uncertainty of their interpretation. But even when that is corrected, the trajectory paths and endpoints by themselves do not identify a smoke-initiation point.

» We agree with the reviewer's point. Trajectories do not by themselves indicate a source origin; rather, they illustrate possible transport pathways conditioned on the meteorological fields. To address this, we have revised wording throughout the manuscript to replace phrases such as "originated from" with "consistent with transport from regions affected by Canadian fires." In addition, our identification of fire source regions was not based on trajectories alone. We used fire location information from the Canadian Interagency Forest Fire Center (via the Canadian Wildland Fire Information System), together with HYSPLIT trajectories, synoptic maps, and satellite imagery, to piece together a consistent picture of source regions and plume transport. This multi-evidence approach avoids over-interpretation of the trajectories as stand-alone "origins".

Section 2.2.6 (2.2.6 Canadian Fire Source Data) is added to the revised manuscript.

2.2.6 Canadian Fire Source Data

We obtained approximate fire source locations from the Canadian Interagency Forest Fire Centre (CIFFC) via the Canadian Wildland Fire Information System (CWFIS) (https://cwfis.cfs.nrcan.gc.ca/). These maps and data products are based on the best available fire reports but may not always reflect the current fire situation in real time. This study employed CIFFC fire locations, HYSPLIT trajectories, synoptic weather maps, and satellite images to find the places that best matched the observed smoke plumes and their transport patterns. This multi-evidence method avoids assigning the term "origin" simply to trajectory analysis.

**Lines 259-265**

[Figure]

For illustration, we attach here a representative CWFIS map for 21 May 2023 showing the active fire locations that informed our analysis. This figure is not included in the manuscript, since it mainly serves to demonstrate the input dataset rather than the results.

The PBL is central to this manuscript. I could not tell from the illustrations where the variable PBL was. Plotting the PBL throughout the lidar time series curtains would be a wonderful addition.

» We agree with the reviewer that the PBL is central to the manuscript and that its depiction in the ceilometer profiles should be more explicit. At present, our captions occasionally rely on qualitative cues (e.g., "PBL is visible as the region with varying depolarization") rather than a

quantitative marker. To address this, we have now computed the PBL height from the CL61 backscatter profile using the Vaisala's proprietary software BL-view and plot it as an overlaid line in each backscatter profile. In this manuscript, references to "PBL height" specifically denote the mixing layer height retrieved from lidar, which we treat as a proxy for the planetary boundary layer height (e.g., Emeis, 2008; Seibert et al., 2000). This addition makes the diurnal evolution of the PBL explicit and removes ambiguity in interpretation.

We appreciate the reviewer's careful reading and practical suggestions. Their points on "descent," the trajectory duration, and the need to show PBL height more clearly pushed us to make several useful corrections/additions to the manuscript. In the revised version of the manuscript, we have softened the terminology, reran the HYSPLIT back trajectories with a global dataset, and added PBL height overlays to the backscatter profiles. These adjustments have improved both the clarity and accuracy of the analysis.

**References:**

Emeis, Stefan, Klaus Schafer, and C. H. R. I. S. T. O. P. H. Munkel. "Surface-based remote sensing of the mixing-layer height-a review." *Meteorologische Zeitschrift* 17, no. 5 (2008): 621.

Seibert, Petra, Frank Beyrich, Sven-Erik Gryning, Sylvain Joffre, Alix Rasmussen, and Philippe Tercier. "Review and intercomparison of operational methods for the determination of the mixing height." *Atmospheric environment* 34, no. 7 (2000): 1001-1027.

Stein, A. F., R. R. Draxler, G. D. Rolph, B. J. B. Stunder, M. D. Cohen, and F. Ngan. "NOAA's HYSPLIT Atmospheric Transport and Dispersion Modeling System", *Bulletin of the American Meteorological Society* 96, 12 (2015): 2059-2077, doi: https://doi.org/10.1175/BAMS-D-14-00110.1

Stohl, Andreas. "Computation, accuracy and applications of trajectories—A review and bibliography." Atmospheric Environment 32, no. 6 (1998): 947-966.

---

## Author Comment (AC3)

Response to RC1:

Reviewer comments in black and response in red.

This manuscript describes analysis of wildfire smoke events over the Baltimore-Washington region using multiple observations and trajectory model simulations. It brings together these multiple lines of evidence for four different events and shows how there can be different transport regimes even for similar satellite signals. The paper is generally well-written and presents interesting results; I think it will be acceptable for publication with relatively minor revisions.

» The authors thank the reviewer for encouraging feedback on our work and for recognizing the need of using multiple observational and modeling tools to evaluate Canadian wildfire smoke events. We appreciate the helpful feedback and will address the individual comments in detail below.

Major Comment: The manuscript focuses on 4 episodes, but contains no discussion of how these were chosen or how these 4 episodes fit into the variations for 2023 (or any other year). Line 91 states "analysing four representative case studies for different plume-PBL interaction scenarios", but I see no discussion of why representative. The Results section starts straight away with the first case study. There needs to be a discussion of how the 4 events were chosen. There also needs to be a discussion of how these episodes compare with the rest of the summer. E.g., Are these the only 4 smoke events (using some criteria)? How does the PM2.5 compare with the rest of the summer? The think plots show a few quantities for each day of the summer would help, both introducing the events and also showing if other events.

» We thank the reviewer for raising this important point. The reason we chose 2023 as the study year is that it was an exceptional wildfire season in Canada, one of the largest on record, with nearly 15 million hectares burned (Jain et al., 2024; Natural Resources Canada, 2024). The smoke from these fires had far reaching implications for health and air quality across North America, with several major cities in the United States of America experiencing unprecedented poor air quality days and severe visibility reductions. These conditions provided strong motivation for us to analyze this year in detail.

Within 2023, the four episodes presented in our manuscript were selected because they represent contrasting transport and mixing scenarios that were clearly captured in the Beltsville dataset. Specifically: (i) a smoke plume apparently lowering into the residual layer followed by frontal

flushing (24-25 May), (ii) a plume showing apparent subsidence under stagnant conditions leading to surface accumulation (5-6 June), (iii) a multilayer intrusion interacting with both residual and convective PBLs (15-17 June), and (iv) a horizontally advected plume already well mixed in the PBL with extreme surface impact (28 June-1 July).

We agree that this rationale was not sufficiently explained in the original draft. In the revised manuscript, we have added a short introductory paragraph at the start of Section 3 clarifying why these cases were chosen and placing them in the broader context of the 2023 summer variability. This addition makes clear that the four events are illustrative, not exhaustive, and were chosen because they represent distinct plume-PBL interaction regimes that could be analyzed with confidence.

Minor Comments.

1. I agree with comments in CC1, especially the length of trajectories shown and the PBL height.

» We have already addressed the trajectory length and PBL height in response to CC1. Specifically, all trajectories have been recomputed using GFS data to the full 72 hr. length and we now overlay PBLH (mixing layer height) estimates on the ceilometer backscatter profiles (see revised Figs. 4, 6, 8, and 10).

2. Line 196. I am not sure "End of May 2023, the …" is correct grammar.

» We agree the original phrasing was incorrect. We have revised "End of May 2023, the …" to "At the end of May 2023, the …" for clarity.
* * *
We sincerely thank the reviewer for constructive feedback and positive assessment of our manuscript. The comments have helped us clarify the rationale behind case selection, improve the consistency of terminology, and strengthen the figures by including full-length trajectories and explicit PBL or mixing layer height overlays. We believe these revisions improve both the clarity and scientific rigor of the paper, and we are grateful for the reviewer's thoughtful input.

---

## Author Comment (AC4)

Response to RC2:

Reviewer comments in black and response in red.

The paper deals with the analyses of different smoke events transported to the Washington – Baltimore area. Intensive fires happened in Canada in late spring 2023 and therefore they are the main study cases, where discussion and analyses try to show the impact of these fires in air-quality for the measurement site in the Howard University Beltsville Campus. Most of the analyses are based on measurements by a Vaisala CL61 Ceilometer capable of providing lidar depolarization and attenuated backscattered at one wavelength. Additional measurements of nitrogen oxides ($NO_x$), carbon monoxide (CO), volatile organic compounds (VOCs), and particulate matter (PM2.5) are presented to support the impact of transported smoke in air-quality. Main analyses are focused on identifying different transport patterns associated with different meteorological conditions and how they ultimately impact particle types in planetary boundary layer (PBL). This objective is of great importance and has potential for its publication in Atmospheric Measurement Techniques. But although the paper is generally well-written, I have some major concerns:

» We thank the reviewer for their careful reading of the manuscript and for recognizing both the importance of the study objective and potential of this work for publication in Atmospheric Measurement Techniques (AMT). We appreciate the acknowledgement of the novelty of the CL61 depolarization measurements in combination with surface air quality observations and trajectory analyses. We also take note of the reviewer's major concerns, which we address carefully and in detail below. In revising the manuscript, we have made clarifications, methodological improvements, and additional context where appropriate, and we believe these changes significantly strengthen the paper.

My first major concern is that authors do not identify appropriately the novelty of this study compared to previous studies in the same region. The Washington – Baltimore area is home of AERONET/Pandora networks, being the Howard University Beltsville Campus less than 15 km from NASA Goddard Space Flight Center where reference instruments of these networks are deployed with long datasets. There are many reference studies using these data (for example, look at Eck et al., 2013). Moreover, the station has been referenced for many multiwavelength lidar Raman developments and there are many studies. For example, Veselovskii et al., (2013) demonstrated how multiwavelength Raman lidar can retrieve aerosol optical and microphysical properties vertically resolved. Fire transport is not new – Veselovskii et al., (2015) study fire transport from California and how it impacts with the PBL. The DISCOVER-AQ field campaign

served to study air-quality and how meteorological conditions and hygroscopic growth ultimately affect aerosol optical and microphysical properties (see Perez-Ramirez et al., 2021). I am discouraged because authors ignore all these studies, and I believe that it must be addressed.

» We thank the reviewer for raising this important point. We agree that the Washington-Baltimore region has been the subject of numerous air quality studies, including pioneering work with AERONET, Pandora, and multiwavelength Raman lidar observations (e.g., Eck et al., 2013; Veselovskii et al., 2013, 2015; Perez-Ramirez et al., 2021). These studies have provided valuable insights into aerosol optical and microphysical properties, hygroscopic growth, and transport of wildfire smoke. We regret that our initial draft did not sufficiently reference these contributions, and we have now revised the Introduction to better situate our study in this context.

The novelty of our work lies not in documenting wildfire transport to the mid-Atlantic for the first time, but in the use of the newly available Vaisala CL61 ceilometer depolarization measurements at the Howard University Beltsville Campus (HUBC) to track wildfire smoke intrusions in near-real time. Unlike the earlier Raman lidar or AERONET/Pandora studies, which typically required either field-campaign deployments (e.g., DISCOVER-AQ) or complex post-processing, the CL61 provides high resolution, continuous, autonomous depolarization profiles. While multiwavelength Raman lidars provide detailed information of the atmospheric tracers, they are expensive and are limited to specialized research facilities or used during short-term campaigns. In contrast, the CL61 being comparatively cost effective, operationally deployable instrument that can complement existing reference sites and extend smoke monitoring into long-term air quality networks. This capability allowed us to distinguish transported smoke from boundary layer aerosols and clouds throughout our study. Our analysis further integrates these vertical depolarization profiles with surface trace-gas and $PM_{2.5}$ measurements, as well as HYSPLIT trajectories, satellite imageries to link lofted smoke layers with their boundary layer mixing and surface air quality impacts. To our knowledge, this is the first of its kind study which systematically uses CL61 depolarization to characterize wildfire smoke intrusions.

Edits made in the revised manuscript Section 1 (Introduction)

**Line # 77-85**, we added citations to Eck et al. (2013), veselovskii et al. (2013, 2015) and Pérez-Ramírez et al. (2021) and acknowledged their contributions.

**Line # 86-87; 92-97**, we now clearly distinguish the novelty of our work

My second major concern is related to the comments from the public discussions, which I support. I did not see clearly how the synoptic conditions changed during the entire study period. Backward-trajectories can give an overview of where air-masses come from, but they alone can

not serve to interpret meteorological conditions. Authors need to give extensive explanations of the meteorological conditions supported by meteorological maps/charts. I encourage homogenizing the explanations as in the current manuscript there are maps for the first event (although only for 500 hPa with is not enough) while there is nothing for the other events.

» We appreciate this comment and agree that synoptic context is essential for interpreting plume transport and plume-PBL interactions. In the original draft, we provided meteorological maps at 500 hPa only for the May case, but we agree this presentation was uneven and insufficient. As the reviewer notes, back trajectories are valuable for illustrating transport pathways, but they must be complemented by synoptic analysis to interpret the dynamical context.

My other major concern is about the use of lidar depolarization as the key parameter to study the transport of smoke particles. Apart from that authors do not specify if it is aerosol depolarization or total depolarization, this parameter only gives information of particle non-sphericity. Lidar community has largely used this parameter only for identifying aerosol non-sphericity (see articles from High Spectral Resolution Lidar developed in NASA Langley). I cannot accept references to Vaisala CL61 White paper (page 15, lines 336 and 339). Authors must acknowledge this limitation and base their study on aerosol attenuated backscatter that is related to aerosol load.

» We appreciate the reviewer's careful point regarding the interpretation of the LDR. We agree that lidar depolarization fundamentally provides information on particle non-sphericity and that it cannot be used as a unique tracer of wildfire smoke. The CL61 reports volume depolarization ratio (VDR), also called the linear depolarization ratio (LDR). It is the cross polarized to co-polarized backscatter ratio without separating aerosol and molecular contributions.

The revised manuscript clarifies that LDR is used only with other evidence, such as aerosol backscatter (aerosol load), surface air quality observations, satellite imagery, and trajectory analysis. We have also removed references to the Vaisala CL61 white paper and now cite peer-reviewed studies that established the use of depolarization for distinguishing spherical and non-spherical aerosols (Tesche et al., 2009; Haarig et al., 2018). In our studies, LDR serves as a supporting discriminator, while attenuated backscatter remains the main indicator of aerosol load.

The novelty of our study lies not in the physics of depolarization itself, but in demonstrating how a commercially available, cost-effective ceilometer with depolarization capability can augment existing satellite and surface datasets to monitor lofted smoke plumes and their interaction with

the boundary layer in near-real time. This work is the first systematic use of CL61 depolarization measurements in the mid-Atlantic region for long-range transported smoke.

Edits made in Section 2.2.1 **Lines 158-168**

Attenuated backscatter profiles are provided as supplementary figures.

My last major concern is that discussion of smoke events is not well-addressed, taking into account the large databases publicly available. Indeed, it is based just in a satellite image. I strongly support the use of satellite aerosol retrievals that can give a quick overview of fires intensity. Also, if authors claim limitations in models they should compare the events with models such as MERRA-2 or CAMS. This has been done in previous studies for the study region (see Veselovskii et al., 2019). The large number of AERONET stations in North America can also serve to have a better picture of these extreme smoke events.

 » We thank the reviewer for highlighting the importance of complementary datasets such as AERONET, MERRA-2, and CAMS. Several studies in the past have used these datasets successfully for this region (e.g., Veselovskii et al., 2019). We chose to emphasize ceilometer LDR and surface chemistry datasets, since our objective was to demonstrate the utility of the newly available CL61 LDR for detecting smoke in the BWC. We used Satellite imagery (VIIRS true color) as a preliminary qualitative confirmation of large-scale plume presence.

All these major concerns should encourage authors to review their conclusions. Particularly, statements such as 'These findings advocate for the wider inclusion of depolarization lidar in regional air-quality networks and the assimilation of near-real-time LDR fields into smoke-forecast systems (Lines 439-440)' or 'Depolarisation lidar proved indispensable for diagnosing smoke layers ' vertical and microphysical character (Lines 448-449).' must be re-thought because LDR only serve for aerosol typing. I insist that for appropriate vertical aerosol optical and microphysical properties characterization multiwavelength lidar Raman are needed, and even these techniques require of case-dependent optimized-constraints (see Perez-Ramirez et al., 2019).

» We agree that the LDR is limited to providing information on particle non-sphericity and cannot on its own yield vertically resolved aerosol optical or microphysical properties. As the reviwer notes, multiwavelength Raman or HSRL systems are established techniques for that

purpose, though even these requires case-specific constraints and retrieval optimization (Pérez-Ramírez et al., 2019).

In our original conclusions, we overstated the role of CL61 depolarization, and we have now revised this language.

At **lines 575 - 577,**
 "These findings advocate for the wider inclusion of depolarization lidar in regional air-quality networks and the assimilation of near-real-time LDR fields into smoke-forecast systems"

has been revised to:

"These findings highlight the potential value of including depolarization-capable, affordable commercial lidars in regional air quality networks and suggest that near-real-time LDR fields could serve as a complementary constraint for smoke forecast systems."

At **lines 599–601,**

"Depolarisation lidar proved indispensable for diagnosing smoke layers' vertical and microphysical character" has been revised to:

"Depolarization lidar provided a useful discriminator for distinguishing lofted smoke layers from boundary-layer aerosols and clouds, but detailed optical and microphysical characterization requires multiwavelength Raman or HSRL observations."

Through these revisions, we want to emphasize that the contribution of CL61 is in aerosol typing and operational monitoring, not in replacing multiwavelength retrievals. We revised the discussion and highlighted that the novelty of our study lies in demonstrating how a cost-effective, autonomous ceilometer with depolarization capability can complement higher-end research instruments and satellite/surface datasets by providing continuous, real-time vertical context for smoke intrusions.

**Minor Concerns**

Lines 44 – 45: Reference needed

» **Line 46** (Bruce et al., 2025) is added as a reference

Line 48: Reference needed

» Noted and changed to highlight that statement belongs to the manuscript's own research " the analysis discussed ahead will show..."

Line 65: Reference needed for the statement of uncertainties of smoke impact on PBL.

» **Line 67** (National Academy of Sciences (2022); Ye et al., 2022) are added as reference

Line 67: Please, specify which past studies.

» Please refer to the citations noted on **Line 72** (Dreessen et al., 2016; Lara et al., 2022)

1.1 – Study region: Give some coordinates for the region

» We added the coordinates of the HU-Beltsville facility in **Line 153-154**

Line 110: Give reference.

» **Line 138 (Manavi, 2025)** is added as a reference

Line 131: LDR can not say anything about aerosol phase state. Please correct.

» Section 2.2.1 is revised, and the above statement is replaced by "The LDR approximates aerosol depolarization in aerosol dominated layers for distinguishing spherical and non-spherical aerosols."

Line 140: How temperature and relative humidity profiles were obtained?

» HUBC performs its own RS-41 radiosonde launches which give a direct measure of Temperature, Relative Humidity, Pressure, Wind Speed and Wind Direction. We have extended the text to clarify the dates and times when launches were made and to show how these correspond to the days of the smoke events.

Figure 1: Please, correct axis. It is difficult to read. Same happens in Figures 4, 6 and 8.

» Thank you for this observation. We have enlarged the font sizes and figure quality is also enhanced.

Revised Figures with large font sizes: Figure 1, 4, 6, and 8.

Figure 1: I observe sub-figures with wind vectors. How were they obtained? Do you have wind profiles? Same happens in Figures 4, 6 and 8.

» The wind vectors shown in Figures 1, 4, 6, and 8 were obtained from measurements collected by a sonic anemometer located at the Maryland Department of the Environment (MDE) site on HU-Beltsville campus. This data provides the near-surface wind speeds and directions used in the corresponding figures.

Line 168: Referenced HYSPLIT appropriately.

» Noted and added.

(Draxler and Hess, 1998; Stein et al., 2015)

2.5: Description of AERONET is vague. Which version of data are you using? What is data quality level used? Also, reference AERONET appropriately.

» Thanks for suggesting expanding the description of AERONET. We have included more detailed description of the program. We have also included a description of the data version and level used to obtain the graphs of Aerosol Optical Depth.

As for the reference, it has been noted and added. (AERONET, 2020).

Figure 3: Why did you use trajectory ensemble? If it provides clear added value. Why did you not use trajectory ensemble for the rest of cases?

» Thank you for pointing that out. We have revised all our HYSPLIT figures to show trajectory ensemble.

Line 238: Figure #? It looks like a error. Same for line 248

» Thank you for pointing this out. Both figures have been identified accordingly.

Line 240: I insist that LDR alone only serves aerosol typing. Be careful.

» Yes, we have corrected throughout the paper that LDR is used with the sole purpose of identifying particle structure. In this particular case, we use LDR to identify non-spherical particles and validating the characterization of these as "smoke" with the use of satellite imagery identifying the aerosol mass originating at the point source of the wildfire events.

Line 245: Why is stabilization of the PBL typical in the evening?

» Thank you for pointing this out. The sentence was not well constructed leading to the confusion.

Revised "The pronounced increase in depolarization ratio within the boundary layer during these periods (ratios reaching up to ~0.1) suggests significant intrusion and mixing of smoke aerosols into the lower troposphere, coinciding with reduction in the PBL turbulence and mixing after sunset as the atmosphere stabilizes." **Lines 332-335**

Lines 257 – 258: Angström exponent can only give information about possible predominance of fine or coarse mode. See studies from AERONET.

» Thank you for your comment. We agree that the Ångström Exponent primarily provides information about the relative predominance of fine versus coarse aerosol modes. In our analysis, we used elevated AE values as supporting evidence that fine-mode aerosols, consistent with smoke particles, were dominant during the intrusion period. We have revised the text to reflect this more accurately and avoid overstating the role of AE in smoke identification.

Revision: "The separation of the 440 nm and 870 nm AOD values, together with elevated AE values, suggests a predominance of fine-mode aerosols, consistent with smoke presence."

Line 264: Give reference for NAQQS limits.

» Noted and added.

Line 281: I do not understand the mention to GOES. Is it a typo?

» Yes, it was a typo and it has been rectified.

Line 301: With your lidar data alone you can not identify particle as fine mode predominance. See my major comments.

» We agree that ceilometer measurements alone cannot definitively identify particle type. In our description, our intention was to highlight correspondence between the observed layer of descent and surface fine particulate concentrations. We have revised the text to clarify that the ceilometer identified the plume's vertical structure and entrainment into the PBL, while the concurrent increase in PM$_{2.5}$ concentrations indicated the presence of fine particulate matter at the surface.

Revision: "From 9 June onward the ceilometer recorded a separate, spectrally distinct return between 3 to 5 km which was identified as the mid-level clouds rather than a smoke plume. Fine particulate concentrations begin to climb after 12 UTC on 6 June, coinciding with the ceilometer's first detection of the plume (shaded) penetrating the PBL (Figure 6). While the ceilometer does not provide particle size information, the simultaneous rise in PM$_{2.5}$ concentrations supports the interpretation that the entrained layer contained fine-mode smoke particles."

Lines 309 – 310: The statement of the impact of clouds in PBL is vague. From your data, is there differences in PBL when these cirrus clouds are present?

» Thank you for this helpful comment. We acknowledge that our original wording on cloud impacts was vague. The ceilometer data during 8 June does not show a strong lowering of the PBL height directly attributable to the mid-level cirrus. However, the presence of these clouds coincided with stagnant conditions and elevated PM$_{2.5}$, suggesting that indirect effects (such as reduced surface heating and modified humidity) may have influenced dispersion. To clarify this point, we have revised the text to more cautiously frame the potential role of clouds.

Revision: "The increase coincided with the ceilometer's cloud-top signature at 3-5 km (Figure 6). While the cloud layer itself was not a particulate source, its presence may have reduced surface heating and modified humidity levels, potentially influencing boundary-layer stability and inhibiting dispersion. However, our data does not show a clear, direct PBL response to the clouds. Disentangling the relative roles of continued smoke advection versus in-situ processes will require chemical speciation and additional trajectory analysis beyond the scope of the present study."

Case study 3: I can not understand the increase of air-quality related parameters If smoke is transported at altitudes above PBL and is decoupled.

» We thank the reviewer for this observation. On 16 June, around 06 UTC, the lidar indicates that the smoke plume began interacting with the top of the PBL rather than remaining fully decoupled

aloft. This interaction does help explain the increases seen in surface $PM_{2.5}$ and $NO_2$. We suppose the reviewer is referring to the elevated ozone values on 15 June. In that case, we agree these were likely not solely from lofted smoke but from a combination of local photochemical production and advected smoke already present within the mixed layer.

Lines 354 – 355: I can not understand your statement of pre-frontal winds role in facilitating vertical mixture.

» Our intent here was to convey that the weak wind conditions ahead of the front limited horizontal advection, allowing the descending smoke layer to remain over the site and gradually mix downward once it intersected the PBL.

Revision: "During the pre-frontal period, weak winds reduced horizontal transport, allowing the descending plume to remain over the site and subsequently mix into the boundary layer."

Lines 358 – 359: AOD is related to aerosol load. For better characterization of smoke properties you need to use AERONET inversion data.

» Thank you so much for your suggestion. The AERONET inversion products do provide better characterization of particle properties like smoke, but for better visualization of data for these cases we decided to display Angstrom Exponent because of its wavelength-dependence of AOD, which is related to aerosol particle size. Finer particles like smoke can be found at Angstrom Exponent values higher than one, supporting the characterization of smoke loading shown with the AOD values in the graph.

Lines 360 – 362: I can not see the impact of smoke on ozone suppression

» We agree that the ozone suppression is not immediately obvious without guidance in the current figure. In the revised manuscript we have added more clarification to assist the readers and the lines the reviewer pointed out to are revised. **Lines 501-507**

Lines 391: Please, provide reference.

» **Line 541** (Asmar et al., 2025) is added as a reference

Line 401 – 402: You have not showed attenuated backscatter that is related to total aerosol load. LDR serves only for aerosol typing. This is related to my previous major comment

» Thank you for pointing this out. We agree that attenuated backscatter is the primary lidar variable related to aerosol load, with depolarization serving as an aid to differentiate between surface aerosols and smoke plumes aloft. In the revised manuscript, we have added attenuated backscatter profiles for each case as Supplementary Figures (S1-S4). The main text and figure captions now explicitly state that backscatter is used to access aerosol abundance. At the same time, LDR is applied only as a supporting discriminator to separate smoke from PBL aerosols and clouds. Reference to the Supplement have been added in Section 3 to guide readers to the corresponding backscatter profiles.

Discussions and Conclusions: I recommend using a table that summarizes the main findings – i.e. statistics of different air-quality related parameters

» Thank you for the suggestion. A compact table can be useful; however, our focus in this research is to illustrate the distinct transport and mixing dynamics through case study analysis. The heterogeneity of the events does not lend itself well to a single statistical table. Instead, we have revised parts of section 4 to highlight the main findings and differences from the four cases. This narrative synthesis achieves the desired clarity without introducing unnecessary redundancy.

**Lines 578-591**

We like to thank the reviewer for evaluation and detailed comments. The concerns raised about novelty, synoptic context, interpretation of depolarization, and balance in the conclusions have led us to make substantial clarifications and improvements. In the revised manuscript, we have better situated our work in the context of prior studies, homogenized the synoptic analysis across all cases, explicitly acknowledged the limitations of depolarization and emphasized attenuated backscatter as the load indicator, and softened the conclusions to highlight CL61 as a complementary and operational tool rather than a replacement for multiwavelength Raman lidar. These revisions have strengthened the manuscript considerably, and we are grateful for the reviewer's input in guiding these improvements.

**References**

Eck et al., 2014: Observations of rapid aerosol optical depth enhancements in the vicinity of polluted cumulus clouds, Chem. Phys., 14, 11633–11656

Perez-Ramirez et al., 2019: Retrievals of aerosol single scattering albedo by multiwavelength lidar measurements: Evaluations with NASA Langley HSRL-2 during discover-AQ field campaigns, Remote Sensing of Environment, 222, 144-164.

Perez-Ramirez et al. 2021: Spatiotemporal changes in aerosol properties by hygroscopic growth and impacts on radiative forcing and heating rates during DISCOVER-AQ 2011, Chem. Phys., 21, 12021–12048, 2021

Veselovskii et al., 2013: Retrieval of spatio-temporal distributions of particle parameters from multiwavelength lidar measurements using the linear estimation technique and comparison with AERONET, Meas. Tech., 6, 2671–2682,

Veslovskii et al., 2015: Characterization of forest fire smoke event nearWashington, DC in summer 2013 with multi-wavelength lidar, Chem. Phys., 15, 1647–1660.